# Serous Tubal Intraepithelial Carcinoma: A Concise Review for the Practicing Pathologist and Clinician

**DOI:** 10.3390/diagnostics10020102

**Published:** 2020-02-13

**Authors:** S. Emily Bachert, Anthony McDowell, Dava Piecoro, Lauren Baldwin Branch

**Affiliations:** 1Department of Pathology and Laboratory Medicine, University of Kentucky College of Medicine, Lexington, KY 40536, USA; emily.bachert@uky.edu (S.E.B.); dava.west@uky.edu (D.P.); 2Department of Obstetrics and Gynecology, Division of Gynecologic Oncology, University of Kentucky College of Medicine, Lexington, KY 40536, USA; amcdowell@uky.edu

**Keywords:** STIC, epithelial ovarian cancer, BRCA mutation

## Abstract

Ovarian cancer is the deadliest gynecologic malignancy, accounting for more than 14,000 deaths each year. With no established way to prevent or screen for it, the vast majority of cases are diagnosed as International Federation of Gynecology and Obstetrics (FIGO) stage III or higher. Individuals with germline *BRCA* mutations are at particularly high risk for epithelial ovarian cancer and have been the subject of many risk-reducing strategies. In the past ten years, studies looking at risk-reducing salpingo-oophorectomy (RRSO) in this population have uncovered an interesting association: up to 8% of women with *BRCA1* or *BRCA2* mutations who underwent RRSO had an associated serous tubal intraepithelial carcinoma (STIC). The importance of this finding is highlighted by the fact that up to 60% of ovarian cancer patients will also have an associated STIC. These studies have led to a paradigm shift that a subset of epithelial ovarian cancer originates not in the ovarian epithelium, but rather in the distal fallopian tube. In response to this, many providers have changed their practice by expanding the role of routine salpingectomy, hysterectomy, and sterilization procedures. The American College of Obstetricians and Gynecologists (ACOG) has acknowledged opportunistic salpingectomy as a safe strategy to reduce the risk of epithelial ovarian cancer in Committee Opinion #774. It is thus important for pathologists and clinicians to understand the definition of STIC; how it is diagnosed; and, most importantly, its clinical significance.

## 1. Background Information

There are three broad categories of primary ovarian tumors: germ cell tumors, sex cord-stromal tumors, and surface/epithelial tumors. Surface epithelial tumors can be subclassified into five different cell types: Brenner, mucinous, clear cell, endometrioid, and serous. In each of these five cell types, tumors can range from benign to borderline (low-malignant potential) to malignant (carcinoma) [1]. Ovarian serous carcinomas can be further divided into low- and high-grade serous carcinomas (HGSOC), the latter being the focus of this review. Women in the United States have a lifetime risk of 1.38% for developing HGSCs, with a mean age of 63 years at presentation. HGSCs have a strong association with *BRCA* mutations and almost ubiquitously harbor *TP53* mutations [2]. Unfortunately, HGSOCs carry a poor prognosis, representing over 70% of all epithelial ovarian cancer deaths [3]. Pathologists and clinicians can thus benefit from a better understanding of tumor pathogenesis. For a multitude of reasons to be discussed in this review, the current prevailing theory is that HGSOCs arise from STICs of the fallopian tube.

## 2. Making the Pathologic Diagnosis

Microscopically, the epithelium of the fallopian tube consists of a mixture of secretory and ciliated cells. The stroma underlying this epithelium is composed of smooth muscle and fibroconnective tissue, and the serosal surface is lined by mesothelium. Interestingly, there is regional variation in the composition of the epithelial layer. Secretory cells predominate in the isthmus of the tube, and ciliated cells are most conspicuous at the fimbriated end [4]. A spectrum of entities originating from secretory cells have been described, including secretory/stem cell outgrowths (SCOUTs), p53 signatures, serous tubal epithelial proliferations or lesions of uncertain significance (STEP-US), and serous tubal intraepithelial carcinomas (STIC). On this spectrum of pathologies, STIC is the most morphologically atypical. It is characterized as secretory cell lesions with some degree of cellular depolarization (typically, but not always, with epithelial stratification), increased nuclear to cytoplasmic ratios, hyperchromasia, nuclear molding, prominent nucleoli, and increased mitotic activity [5,6] (Figure 1). In addition to these morphologic criteria, *TP53* mutations are present in 92% of STICs. The lesions therefore typically demonstrate strong and diffuse p53 immunohistochemical staining consistent with missense mutations (Figure 1B). Less commonly, there is complete absence of staining due to nonsense mutations in *TP53* [2].

As morphologic findings may sometimes be subtle, interpretations are unavoidably subjective. It is thus not unexpected that there is poor reproducibility in diagnosing these precursor lesions. At least two studies have confirmed this low interobserver agreement. First, Carslon et al. circulated digital images of 30 cases (14 cases of STIC and 16 cases with benign tubal epithelium) to 12 pathologists. The majority agreed on STIC in only nine of 14 cases, resulting in a κ statistic of 0.333 [7]. A second study by Visvanathan et al. highlighted that even intraobserver variability for diagnosing STIC versus non-STIC was lackluster based on morphologic findings alone (κ ranged from 0.41 to 0.68 for five pathologists) [6]. As such, Vang et al. validated an algorithm using an on-line training set of H&E and immunohistochemical stained slides (p53 and Ki-67) originally developed by Visvanathan et al. [6,8]. In these studies, foci morphologically suspicious for STIC needed to show a Ki-67 labeling index >10% and a mutant p53 staining pattern to be diagnosed as such. The group was able to increase the interobserver κ value for STIC vs. non-STIC to 0.67 by combining the morphologic features with p53 and Ki-67 immunostaining results [8]. While this is an improvement, there is clearly still substantial discordance. To appropriately diagnose STIC, pathologists should liberally use p53 and Ki-67 immunostains as well as seek second opinions from colleagues.

STICs are not identifiable grossly. As such, the Sectioning and Extensively Examining the Fimbriated End (SEE-FIM) protocol was published in 2006 in an effort to detect STICs more reliably [9]. The protocol calls for prophylactic salpingo-oophorectomy specimens to be submitted in their entirety. For the fallopian tubes, the distal two centimeters, including the fimbria, are amputated and then longitudinally sectioned into four sections. The reminder of the fallopian tube is sectioned at 2 mm to 3 mm intervals [9]. Classical grossing procedures typically included only three cross-sections of fallopian tube. It was felt that the SEE-FIM protocol would be of particular use to the *BRCA*-positive women whose management is more likely to be impacted by the finding of a tubal malignancy. Furthermore, an “incidental” STIC discovery may lead to identifying *BRCA*-positive patients and possibly benefit their family members.

The theory that this protocol would increase detection of STIC was corroborated early on by a small but illustrative study by Medeiros et al. The authors evaluated a series of bilateral salpingo-oophorectomy specimens from both *BRCA*-positive patients and non-high risk patients using the SEE-FIM protocol. Of the 26 cases (13 in each group), five tumors were identified, four of which were at the fimbriated end. The fifth was located in the ampullary segment of the fallopian tube [10].

Furthermore, a recent study compared the efficacy of the SEE-FIM protocol versus classical grossing methods in detecting various microscopic lesions, including STICs. From 536 cases grossed by the SEE-FIM protocol, 15 STICs were identified in 39 cases of non-uterine pelvic high-grade serous carcinoma. Most cases of STIC were located at the fimbriated end of the fallopian tube. Of 582 cases evaluated by the classical method, only one STIC was identified among patients with non-uterine pelvic HGSC [11].

## 3. Pathogenesis of HGSOCs and the Role of STICs

It is currently accepted that STICs are a precursor lesion to ovarian HGSCs, a theory proposed in the late 1990s [2,12,13]. The strongest evidence for this theory is the observance of identical somatic *TP53* mutations in STICs and concurrent pelvic HGSCs. Kuhn et al. demonstrated identical mutations in 27 of 29 paired cases of STICs and HGSCs, including missense mutations in 61% of cases and null mutations (frameshift/splicing junction/nonsense mutations) in 39% [14]. Furthermore, their study also went on to show that immunohistochemistry for p53 can be used as a reliable surrogate for identifying a *TP53* mutation (with 87% sensitivity and 100% specificity). Missense mutations result in strong, diffuse staining for p53, while the null mutations show complete loss of staining. Wild-type staining pattern (negative *TP53* mutation) has weak, patchy nuclear staining [14].

Moreover, the same team also established that in cases of paired STICs and concurrent HGSOCs, the STICs have shortened telomere lengths compared to normal tubal epithelium, and the HGSOCs have longer telomeres than their counterpart STICs. STICs were postulated to represent precursors to ovarian HGSOCs (as opposed to metastases) because telomere shortening is one of the known early events in the transition to neoplasia. Subsequent lengthening/stabilization of telomeres is likely necessary to support the rapid cell division seen in most carcinomas [15].

A recent systematic literature review by Chen et al. demonstrated that STICs coexist with high-grade serous ovarian carcinomas in a mean of 31% of cases (range: 11–61%, 95% CI: 17–46%). Two of the reviewed studies excluded patients with known *BRCA* mutations. The rate of co-existent STIC and HGSC was 33% and 66% in patients without these mutations [16]. More recently, Ducie et al. showed there was no significant differences between HGSCs with and without STIC at the time of diagnosis with regard to copy number alterations, messenger RNA sequence, and microRNA profile [17].

Obviously, not all pelvic high-grade serous carcinomas are associated with STICs. This issue is further complicated by the fact that HGSCs often present at such an advanced stage that the fallopian tube is obliterated. As such, there remain alternative hypotheses for the origin of these malignancies. One alternative posits that ovarian surface epithelium or inclusion cysts in the ovarian cortex are precursors. This was the predominant theory before the role of STICs was identified. This theory is logical given that the outer ovarian surface is the most common location for these tumors. However, the lack of *TP53* mutations in these cysts argues against this option [18]. An additional alternative is the “secondary Müllerian system” (Müllerian remnants on peritoneal surfaces including endosalpingiosis and endometriosis) as the source of some serous ovarian carcinomas. However, as with the former proposal, these remnants rarely express aberrant p53 expression [18].

Each of these proposals has flaws, and as such, several unifying theories have been recently proposed:

(1) The “precursor escape” model by Soong et al. states that the early secretory cell proliferations, p53 signatures, can function as a distinct precursor to HGSOCs rather than as just being precursors to STICs. These cells with *TP53* mutations can detach from the fallopian tube and can undergo malignant transformation in the peritoneal cavity, thereby creating a HGSOC with no obvious site of origin [18].

A recent study by Wu et al. performed whole-exome sequencing on fallopian tube precursors in women with and without concurrent ovarian high-grade serous carcinoma to help elucidate the early molecular events of tumorigenesis. A total of 18 precursor lesions from patients without germline *BRCA* mutations was sequenced after isolation by laser capture microscopy. The average number of mutations in p53 signatures was less than that in STILs and in STICs, although it did not reach statistical significance. Furthermore, incidental STICs had fewer somatic mutations than those STICs with concurrent ovarian high-grade serous carcinomas (*p* = 0.01). Their work supports the notion of the progression of p53 signatures to STICs [19].

(2) Banet and Kurman offer a somewhat unique hypothesis: that some cortical inclusion cysts of the ovary arise from tubal epithelium implanting onto the ovarian surface when disrupted during ovulation. The authors evaluated a cohort of 35 patients with incidental ovarian cortical inclusion cysts at autopsy and found that 60% were lined by ciliated, tubal-type columnar epithelium, 14% with flat epithelium, and 31% with both. Regarding immunohistochemistry for PAX-8 and calretinin, the ciliated epithelial cysts were consistent with tubal origin (positive for PAX-8) and the flat epithelial cysts were consistent with cells of mesothelial origin (positive for calretinin). Furthermore, no ciliated cysts were found in patients younger than 12 years of age, supporting their tubal origin with ovulation theory [20].

(3) In a sophisticated study by Zhang et al., genetically engineered mouse models and organoids were used to demonstrate that both candidate sites (fallopian tube epithelium and ovarian surface epithelium) are cells of origin for serous ovarian carcinomas. Importantly, their findings also suggested that the cell of origin may influence response to chemotherapy. Fallopian tube epithelium-derived organoids were more likely to be sensitive to paclitaxel and carboplatin compared to ovarian surface epithelium-derived organoids, with statistical significance. Both lines demonstrated similar sensitivity to gemcitabine, niriparib, and olaparib [21].

Regardless of our gaps in knowledge, it is almost certain that STICs play an important contributing role to the pathogenesis of HGSOCs. Therefore, pathologists should remain diligent in recognizing and reporting these precursor lesions. Furthermore, pathologists should remain diligent in properly classifying the primary site for HGSOCs. Practically speaking, the only situation in which the ovary should be classified as the primary tumor site is in the presence of an ovarian mass with no STIC or tubal involvement in either fallopian tube. Fallopian tubes should be extensively sampled per the SEE-FIM protocol [22,23].

## 4. Association with Other Malignancies

The association of STIC with endometrial malignancies has been evaluated by rare studies and remains debatable, at best. Tolcher et al. looked at 38 cases of uterine serous carcinomas. Of these cases, 11 demonstrated some form of fallopian tube involvement. Only two of the 11 cases demonstrated STIC. Interestingly, both of the STIC lesions were in cases with stage III or IV uterine serous carcinomas. The team concluded that a small minority of uterine serous carcinomas may be attributable to a primary origin in the fallopian tube. However, as the authors pointed out, it is more logical that these STICS are metastatic implants given the higher stage of these uterine carcinomas and the lack of tubal invasion [24]. Kommoss et al. performed a similar retrospective study of 161 uterine serous carcinoma cases. Thirty-two cases demonstrated tubal involvement, and 17 of these showed STIC features. Once again, the tubal involvement was characterized as metastatic in the vast majority of cases [25].

An older study by Tang et al. demonstrated similar findings with 4/28 cases (14%) of endometrial serous carcinoma co-existing with STIC. Seventy-four additional endometrial non-serous malignancies (predominantly endometrioid type) were negative for co-existing STIC [26]. Again, whether these represent synchronous multifocal lesions, metastases, or site primary origin is unknown. Whether the presence of co-existent STIC has any prognostic significance has also not been systematically evaluated.

## 5. Prevalence

One of the first studies to link *BRCA* germline mutations with STICs was published back in 2000, which demonstrated “fallopian tube cancer” in two patients with *BRCA1* germline mutations who also had loss of the wild-type *BRCA1* allele in the tumor tissue [27]. Since this time, knowledge has exploded regarding the overall prevalence of STICs and their strong association with *BRCA* mutations. It is known that *BRCA1* carriers have a 44% risk of developing ovarian cancer, and there is a 17% risk for *BRCA2* carriers [28]. As such, given the prevailing theories of the role of STIC, it is only logical to conclude these women are also at a significantly increased risk of developing STIC.

Most recently, a retrospective study of 527 patients over a span of 16 years evaluated *BRCA1/2* carriers undergoing risk-reducing salpingo-oophorectomy. HGSOCs were identified in 2.3% of patients and 59%, were classified as of fallopian tube origin (per aforementioned criteria by Singh) [23,29]. The pith of the study, however, was the finding of isolated STIC in 0.8% of women (4/527). Of these isolated STICs, two patients with *BRCA1* germline mutations went on to develop peritoneal serous carcinoma after a prolonged interval of >7 years. Clonality of the two lesions was established by next generation targeted sequencing demonstrating identical *TP53* mutations [29]. Clearly, this long intervening period causes improper clinical follow-up of these patients. Another study, a 2015 systematic retrospective literature review by Patrono et al., found a 4.5% rate of developing primary peritoneal carcinoma in patients with an isolated STIC and *BRCA* mutations. Again, there was no standard treatment plan with regards to surgery or adjuvant therapy for these patients [30].

A large multicenter study also evaluated high-risk women (92% with a *BRCA1/2* mutation) who underwent risk-reducing salpingo-oophorectomy. Using extensive sampling, STICs or STILs were identified in 11.9% of women, p53 signatures in 27.0%, and more than one lesion in 50%. Of note, tissue blocks of fallopian tube fimbria were flipped and sectioned to allow for more extensive tissue evaluation. Per the authors, this resulted in twice as many lesions being reported [31]. Of course, such extensive sectioning of tissue blocks is not routine practice.

On the other hand, for the general population (women without known *BRCA1/2* mutations), so called “incidental” STICs remain quite rare. Meserve et al. reported a single institution case review of bilateral salpingectomies performed in women over 50 years of age. Of 1747 bilateral salpingectomies performed in women with no known risk factors, only two cases of STIC were diagnosed. Of these two cases, one was associated with an ovarian low grade serous carcinoma and the other with a grade 2 endometrioid-type endometrial carcinoma. While the team did point out that grossing of the specimens was not entirely uniform, it can be concluded that prevalence of “incidental” STICs is low. This brings to question the value of so-called opportunistic salpingectomies [32].

## 6. Clinical Prevention and Detection

Even so, ACOG has recommended opportunistic salpingectomy as a primary prevention of ovarian carcinoma in females undergoing pelvic surgery for other indications or as a sterilization procedure. ACOG cited a large Swedish population-based cohort study that reported a risk reduction of ovarian cancer in females who had a bilateral salpingectomy versus others (hazard ratio of 0.65) [33]. Furthermore, if complete salpingectomy (from the fimbriated end up to the uterotubal junction) is unable to be performed, they recommend removing as much of the tube as feasibly possible [34]. For BRCA patients, the role of prophylactic salpingo-oophorectomy is clearly stated. In ACOG Practice Bulletin 182: “risk-reducing salpingo-oophorectomy is recommended at age 35–40 years for *BRCA1* carriers with the highest lifetime risk of ovarian cancer, whereas women with *BRCA2* may consider delaying until age 40–45 years because of later onset of ovarian cancer.” It is less clear if there is a role of staged salpingectomy followed by oophorectomy to avoid the morbidity of early menopause. This exact question is currently being investigated in the TUBA study [35].

Wong et al. evaluated 66 subjects who underwent salpingectomy immediately followed by oophorectomy, for a total of 107 ovaries examined. The patients did not have a history of ovarian pathology and were undergoing the procedure for risk reduction. Residual salpingeal tissue was observed in only 6 of the 107 ovaries (5.6%) [36]. Unfortunately, given the design of the study, there was no long-term follow-up to determine if the presence or volume of salpingeal tissue left behind correlated with development of a pathologic lesion.

As noted previously, early diagnosis remains elusive for HGSCs. The US Preventive Services Task Force (USPSTF) recommends against screening for ovarian cancer in asymptomatic women, with a grade D recommendation. Current screening options consist only of transvaginal ultrasound and/or serum CA-125 levels. Both of these have proved to be ineffective. In addition, ACOG, the American Cancer Society, and the American College of Radiology all do not recommend screening in the average-risk woman [37]. As such, major work still needs to be done to help identify patients at risk without known *BRCA1/2* mutations.

A panel of methylation biomarkers has been recently suggested as a method to diagnose HGSCs not only early in their clinical presentation, but also at the precursor stage. The hypermethylation of three genes (*c17orf64, IRX2,* and *TUBB6*) was able to discriminate HGSC with a sensitivity and specificity of 100% from control tissues. In addition, they were also found in fallopian tubes with STIC, but not in pathologically unremarkable fallopian tubes [38]. While these findings need to be corroborated by other studies before such biomarkers can be of clinical use, these results offer promise in the quest for early cancer detection.

## 7. Conclusions

As our ability to diagnose STIC has improved, so has our understanding of its association with HGSOCs and *BRCA* mutation carriers. This has complicated clinical decision making. Namely, what should clinicians do when an isolated STIC is diagnosed? There is currently no clear guideline on postsurgical treatment after incidental STIC is found. As we move forward with advancing our understanding of the clinical-pathologic associations of this lesion, it is imperative that we carefully incorporate this finding into future clinical algorithms.

In summary, for pathologists, consistent and thorough sampling of fallopian tubes with a low threshold of immunohistochemistry will help diagnose STIC, thus identifying potential at-risk patients and their families. For clinicians, adequate prolonged follow-up for incidental STIC with referral to Gynecologic Oncology and encouragement of opportunistic salpingectomy will remain a mainstay. The role of considering STIC in staging can be highlighted by the ESMO-ESGO guidelines, but treatment recommendations remain inconclusive.

## Figures and Tables

**Figure 1 diagnostics-10-00102-f001:**
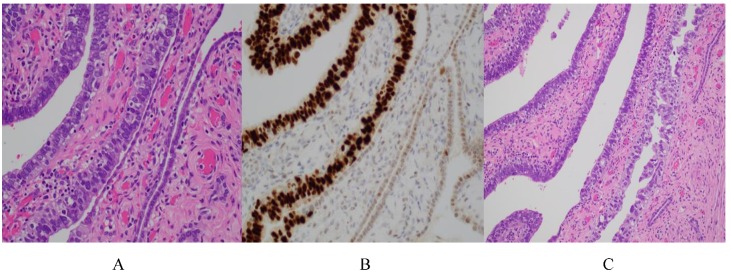
(**A**) Serous tubal intraepithelial carcinoma (STIC) (left) versus uninvolved fallopian tube (right). The STIC demonstrates loss of polarity with an increase in nuclear size with nuclear crowding and molding. (**B**) p53 immunohistochemistry highlighting strong and diffuse expression in STIC (left) versus uninvolved fallopian tube (right). (**C**) Lower magnification highlighting the hyperchromasia and marked nuclear pleomorphism associated with STIC.

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
