# Peer review of "Serous Tubal Intraepithelial Carcinoma: A Concise Review for the Practicing Pathologist and Clinician"

_diagnostics, 2020, doi:10.3390/diagnostics10020102_

Round 1

Reviewer 1 Report

Interesting topic and review.  Not completely comprehensive.

Several citations could still have been included:  additional citations from Kurman, Koh, Vang, Chene might have been included.

No mention made of the possible utility of Ki67 in the determination of STIC.

Author Response

Additional citations and mention of Ki67 were included. Further attempt to make this a comprehensive review were made with emphasis to include clinically relevant material.

Reviewer 2 Report

This is a very short review article with only 15 references cited.  Among these 15 references, only two were published in 2019 and one of this two references is also a review. In the last three years, there are over 70 papers about STIC lesions. What type of literature search was conducted and over what time periods for this review are not clearly defined.

This review is not written comprehensive and scholarly. There is no subheadings and tables to organize, synthesize and explicate findings including inconsistent findings. In the abstract and the first paragraph, it mentioned that "there is still much to be clarified.", but not much was discussed in the subsequent paragraphs. The review also missed one of the most important article from the ESMO-ESGO consensus conference (Annals of Oncology, Volume 30, Issue 5, May 2019, Pages 672–705) with recommendations related to the presence of STIC such as:

Recommendation 1.7: STIC should count as a disease site for staging purposes; for example, a case with a STIC and HGSC confined to the ovary should be staged as stage IIA fallopian tube HGSC.

Recommendation 6.5: peritoneal restaging should be considered in cases of incidentally detected, apparently isolated STIC lesions.

Recommendation 8.2: adjuvant chemotherapy is not recommended in the management of incidentally detected isolated STIC lesions.

Author Response

The authors appreciate your honest review. The thorough and detailed comments were considered in the revisions and, hopefully, adequately reflect the suggestions you made. With respect to the ESMO guidelines, the authors are more accustomed to the NCCN guidelines. Furthermore, those recommendations are mostly level 3 and 4 evidence. The guideline itself makes the point that "there continues to be disagreement on primary site assignment." Regardless, the revised paper hopefully does a better job at emphasizing the relationship between HGSC and STIC.

Reviewer 3 Report

In the manuscript submitted by Bachert et al., reviews the recent advance on serous tubal intraepithelial carcinoma (STIC), the presumed precursor lesion of ovarian high-grade serous carcinoma. The article is, in general, well written but the information provided is limited, especially at the molecular genetic aspects. For a review article, the references cited are insufficient and do not cover many recent major findings in this field.  The manuscript can be recommended for publication if it is adequately revised.

Major comments

The author should mention the histological criteria for STIC and other tubal precursors in the beginning of the discussion so the authors have a general ideal what is a STIC (PMID: 22498942, 21989347).

The prevalence of tubal lesions including STIC in a multicenter study should be described (PMID: 30232083).

In addition to TP53 mutation, there are a couple of alterations shared between STIC and HGSOC could support the hypothesis that fallopian tube is the primary site of high-grade pelvic serous carcinoma. These include shortened telomere (31107721, 23589176), overexpression of laminin C(22892598), cyclin E, fatty acid synthase, rsf-1(20228782), STMN1, TET1(30883733), and loss of PAX2(20597068), FOXO3a(24077281) and PTEN(20562848). The underlying mechanisms of these alterations and their role in disease progression are still under investigation.

The author should mention the molecular alterations in STIC discovered by three recent studies performing WES (for example, 30560554) and one methylome profiling (30108103) on tubal lesions. These publications represent the major studies on the molecular genetic and epigenetic aspects of STICs.

Based on somatic mutations and LOH, the two recent studies performing phylogenic analysis on STIC and its potential precursors. It is estimated progression from STIC to ovarian HGSC takes ~6 years whereas from HGSOC to metastases takes only 2 years. This warrants new diagnostic strategy to early detect HGSOC when at preinvasive stage (30560554, 29061967). This important information should be discussed.

Minor comments

Line 99-103: it’s better to use HGSOC instead of HGSC as the purpose of this paragraph is to explain why 40% of HGSOC don’t have associated STIC. The early serous proliferation (ESP) in the original article include p53 signature and STIL, not just p53 signature (30043522).

An alternative explanation for 40% of HGSOC patients don’t have an associated STIC is the hypothesis that normal fallopian tube epithelium implanting at the site of ovarian surface rupture during ovulation, leading to the development of cortical inclusion cyst that can then undergo malignant transformation (20436685).

Line 135-136: “However, the mutational underscoring of STIC and the p53 signature has clouded these presumed associations and has complicated clinical decision making.” The meaning of this sentence is unclear. p53 signature should also be removed from this sentence as its high frequency in general population and is not considered as premalignant.

In the last paragraph, given the window of time between fallopian tube lesion and development of HGSOC, the author should mention the recent development of new screening tools including PapGene and PapSEEK to detect STIC as an important future direction.

Since the journal is not a specialty oriented one, it would be useful to give a general introduction of different types of ovarian cancer in the first paragraph then give a reason to focus on high-grade serous carcinoma and STIC. A recent review on this topic can be cited: Am J Pathol Volume 186, 2016, Pages 733-747.

Author Response

The authors would like to thank you for the time you took to provide the major revisions.  Each point was considered and hopefully adequately reflected in the changes that we made. It is difficult to concisely dress each point because the entire paper has been restructured and formatted to appropriately.

Round 2

Reviewer 1 Report

Much improved organization and review of the literature.  Be sure to spell check again.

Reviewer 2 Report

The revised review has made significant improvement with subheadings and two additional 2019 reference.

Reviewer 3 Report

None